# Measuring and mitigating interference in reinforcement learning

## Abstract

Catastrophic interference is common in many network-based learning systems, and many proposals exist for mitigating it. Before overcoming interference we must understand it better. In this work, we provide a definition and novel measure of interference for value-based reinforcement learning methods such as Fitted Q-Iteration and DQN. We systematically evaluate our measure of interference, showing that it correlates with instability in control performance, across a variety of network architectures. Our new interference measure allows us to ask novel scientific questions about commonly used deep learning architectures and study learning algorithms which mitigate interference. Lastly, we outline a class of algorithms which we call online-aware that are designed to mitigate interference, and show they do reduce interference according to our measure and that they improve stability and performance in several classic control environments.

## 1 Introduction

A successful reinforcement learning (RL) agent must generalize — learn from one part of the state space to behave well in another. Generalization not only makes learning more efficient but is also essential for RL problems with large state spaces. For such problems, an agent does not have the capacity to individually represent every state and must rely on a function approximator — such as a neural network — to generalize its knowledge across many states. While generalization can improve learning by allowing an agent to make accurate predictions in new states, learning predictions of new states can also lead to inaccurate predictions in unseen or even previously seen states. If the agent attempts to generalize across two states that require vastly different behavior, learning in one state can interfere with the knowledge of the other. This phenomenon is commonly called *interference*[1] or forgetting in RL (Bengio et al., 2020; Goodrich, 2015; Liu et al., 2019; Kirkpatrick et al., 2017; Riemer et al., 2018).

The conventional wisdom is that interference is particularly problematic in RL, even single-task RL, because (a) when an agent explores, it processes a sequence of observations, which are likely to be temporally correlated; (b) the agent continually changes its policy, changing the distribution of samples over time; and (c) most algorithms use bootstrap targets (as in temporal difference learning), making the update targets non-stationary. All of these issues are related to having data and targets that are not iid. When learning from a stream of temporally correlated data, as in RL, the learner might fit the learned function to recent data and potentially overwrite previous learning—for example, the estimated values.

To better contextualize the impacts of interference on single task RL, consider a tiny two room gridworld problem shown in Figure 1. In the first room, the optimal policy would navigate to the bottom-right as fast as possible, starting from the top-left. In the second room, the optimal policy is the opposite: navigating to the top-left as fast as possible, starting from the bottom-right. The agent is given its position in the room and the room ID number, thus the problem is fully observable. However, the agent has no control over which room it operates in. We can see catastrophic interference if we train a DQN agent in room one for a while

---

[1]The term *interference* comes from early work in neural networks (McCloskey & Cohen, 1989; French, 1993; 1999). McCloskey and French (McCloskey & Cohen, 1989) showed that neural networks, when trained on successive supervised learning tasks using gradient descent, over-wrote the knowledge of earlier tasks with that of newer tasks.

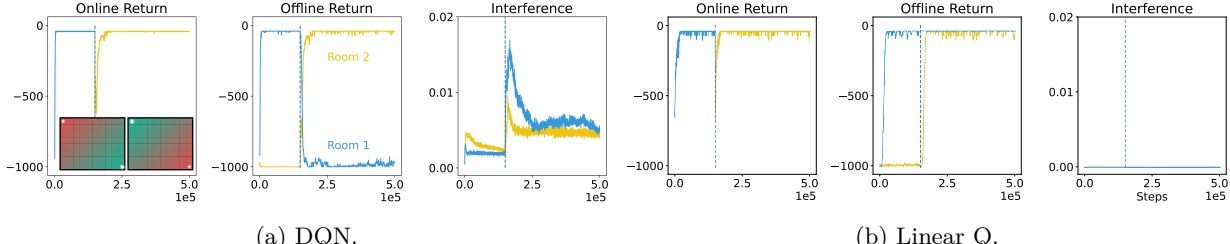

(a) DQN.                                                                    (b) Linear Q.

Figure 1: The Two-Room environment: a simple diagnostic MDP to highlight the impacts of interference. The inset of the diagram above shows how the two rooms of the environment are not connected. The optimal policy and value function are opposite in each room as indicated by the color gradient. Each room is a grid of size $20 \times 20$. One third of the way through training the agent is teleported from room 1 to room 2. A DQN agent experiences significant interference upon switching to the second room negatively impacting its performance in the first room (as shown by offline return), even though the observation enables perfect disambiguation of the rooms.

and then move the agent to room two. The agent simply overrides its knowledge of the values for room one the longer it trains in room two. Indeed, we see DQN's performance in room one completely collapse. In this case the interference is caused by DQN's neural network. We contrast this to a simple tile coding representation (fixed basis), with a linear Q-learning agent. The tile coding represents these two rooms with completely separate features; as a result, there is no interference and performance in room one remains high even when learning in room two.

It is difficult to verify this conventional wisdom in more complex settings, as there is no established online measure of interference for RL. There has been significant progress quantifying interference in supervised learning (Chaudhry et al., 2018; Fort et al., 2019; Kemker et al., 2018; Riemer et al., 2018), with some empirical work even correlating interference and properties of task sequences (Nguyen et al., 2019), and investigations into (un)forgettable examples in classification (Toneva et al., 2019). In RL, recent efforts have focused on generalization and transfer, rather than characterizing or measuring interference. Learning on new environments often results in drops in performance on previously learned environments (Farebrother et al., 2018; Packer et al., 2018; Rajeswaran et al., 2017; Cobbe et al., 2018). DQN-based agents can hit performance plateaus in Atari, presumably due to interference. In fact, if the learning process is segmented in the right way, the interference can be more precisely characterized with TD errors across different game contexts (Fedus et al., 2020). Unfortunately this analysis cannot be done online as learning progresses. Finally, recent work investigated several different possible measures of interference, but did not land on a clear measure (Bengio et al., 2020).

Interference classically refers to an update negatively impacting the agent's previous learning—eroding the agent's knowledge stored in the value function. Therefore it makes sense to first characterize interference in the value function updates, instead of the policy or return. In most systems the value estimates and actions change on every time-step conflating many different sources of non-stationarity, stochasticity, and error. If an update to the value function interferes, the result of that update might not manifest in the policy's performance for several time steps, if at all. We therefore focus on measuring interference for approximate policy iteration algorithms: those that fix the policy for some number of steps (an iteration) and only update the value estimates.

We specifically conduct experiments on a class of algorithms we call Deep Q-iteration. One instance—with target networks—is almost the same as DQN but additionally keeps the behavior policy fixed within an iteration. The goal is to remove as many confounding factors as possible to make progress on understanding interference in RL. This class of algorithms allows us to investigate an algorithm similar to DQN; investigate the utility of target networks; and define a sensible interference measure by keeping more factors of variation constant within an iteration.

The contributions in this work are as follows. (1) We define interference at different granularities to capture interference within and across iterations for this class of value-based algorithms. (2) We justify using differences in squared TD errors across states before and after an update as an effective and computationally efficient approximation of this interference definition. (3) We empirically verify the utility of our interference metric by showing that it correlates with instability in control performance across architectures and optimization choices. (4) We leverage this easy-to-compute measure to outline a class of algorithms that mitigate interference. We demonstrate that these *online-aware* algorithms can improve stability in control by minimizing the interference metric. We conclude this work by highlighting limitations and important next steps.

## 2 Problem Formulation

In reinforcement learning (RL), an agent interacts with its environment, receiving observations and selecting actions to maximize a reward signal. We assume the environment can be formalized as a Markov decision process (MDP). An MDP is a tuple $(\mathcal{S}, \mathcal{A}, \Pr, R, \gamma)$ where $\mathcal{S}$ is a set of states, $\mathcal{A}$ is an set of actions, $\Pr : \mathcal{S} \times \mathcal{A} \times \mathcal{S} \to [0,1]$ is the transition probability, $R : \mathcal{S} \times \mathcal{A} \times \mathcal{S} \to \mathbb{R}$ is the reward function, $\gamma \in [0,1)$ a discount factor. The goal of the agent is to find a policy $\pi : \mathcal{S} \times \mathcal{A} \to [0,1]$ to maximize the expected discounted sum of rewards.

Value-based methods find this policy using an approximate policy iteration (API) approach, where the agent iteratively estimates the action-values for the current policy and then greedifies. The action-value function $Q^{\pi} : \mathcal{S} \times \mathcal{A} \to \mathbb{R}$ for policy $\pi$ is $Q^{\pi}(s,a) := \mathbb{E}[\sum_{k=0}^{\infty} \gamma^k R_{t+k+1} | S_t = s, A_t = a]$, where $R_{t+1} := R(S_t, A_t, S_{t+1})$, $S_{t+1} \sim \Pr(\cdot|S_t, A_t)$, and $A_t \sim \pi(\cdot|S_t)$. The Bellman operator for action values $\mathcal{T}^{\pi} : \mathbb{R}^{|\mathcal{S}| \times |\mathcal{A}|} \to \mathbb{R}^{|\mathcal{S}| \times |\mathcal{A}|}$ is defined $(\mathcal{T}^{\pi}Q)(s,a) := \sum_{s' \in \mathcal{S}} \Pr(s'|s,a) [R(s,a,s') + \gamma \sum_{a' \in \mathcal{A}} \pi(a'|s')Q(s',a')]$. This operator can be used to obtain $Q^{\pi}$ because it is the unique solution of the Bellman equation: $\mathcal{T}^{\pi}Q^{\pi} = Q^{\pi}$. Temporal difference (TD) learning algorithms are built on this operator, as the sampled TD error $\delta$ in expectation equals $\mathcal{T}^{\pi}Q - Q$. We can use neural networks to learn an approximation $Q_{\boldsymbol{\theta}}$ to the action-values, with parameters $\boldsymbol{\theta}$. Under certain conditions, the API procedure—consisting of a policy evaluation step to get $Q_{\boldsymbol{\theta}}$, followed by greedifying to get a new policy, and repeating—eventually converges to a nearly optimal value function (Sutton & Barto, 2018).

We investigate a particular API algorithm that is similar to Deep Q-learning (DQN), that we call Deep Q-iteration. The only difference to DQN is that the behavior policy is held fixed during each evaluation phase. In this algorithms there is an explicit evaluation phase for a fixed target policy, where the agent has several steps $T_{\text{eval}}$ to improve its value estimates. More specifically, on iteration $k$ with current action-values estimates $Q_k$, the target policy is greedy $\pi_k(s) = \arg\max_{a \in \mathcal{A}} Q_k(s,a)$ and the behavior is $\epsilon$-greedy. For each step in the iteration, a mini-batch update from a replay buffer is performed, using the update equation

$$\Delta\theta := \delta_t \nabla_{\boldsymbol{\theta}_t} Q_{\boldsymbol{\theta}_t}(S_t, A_t)$$

for temporal difference (TD) error $\delta_t$. This TD error can either be computed without target networks, $\delta_t := R_{t+1} + \gamma Q_{\boldsymbol{\theta}_t}(S_{t+1}, \pi_k(S_{t+1})) - Q_{\boldsymbol{\theta}_t}(S_t, A_t)$, or with a target network, $\delta_t = R_{t+1} + \gamma Q_k(S_{t+1}, \pi_k(S_{t+1})) - Q_{\boldsymbol{\theta}_t}(S_t, A_t)$. The procedure for is summarized in Algorithm 1.

We exactly recover DQN by setting the behavior policy[2] to be $\epsilon$-greedy in $Q_{\boldsymbol{\theta}_t}$ rather than $Q_k$. We opt to analyze this slightly modified algorithm, Deep Q-iteration, to avoid confounding factors due to the policy changing at each step. The definitions of interference developed in the next section, however, directly apply to DQN as well. For the controlled Two Rooms example in the introduction, we used our measure for a DQN agent. However, when moving to more complex, less controlled scenarios, this changing data distribution may impact outcomes in unexpected ways. Therefore, to control for this factor, we focus in this work on Deep Q-iteration algorithms where the data-gathering policy also remains fixed during each iteration.

---

[2]Notice that the typical bootstrap target $\max_{a'} Q_k(S_{t+1}, a')$ in DQN is in fact equivalent to the Deep Q-iteration update with a target network, because $\max_{a'} Q_k(S_{t+1}, a') = Q_k(S_{t+1}, \pi_k(S_{t+1}))$. The scalar $T_{\text{eval}}$ is the target network refresh frequency. We can also recover Double DQN (Hasselt et al., 2016), though it deviates just a bit more from Deep Q-iteration. It similarly uses the Deep Q-iteration update with a target network, but the target policy is greedy in $Q_{\boldsymbol{\theta}_t}$ rather than $Q_k$. The resulting TD error is instead $\delta_t := R_{t+1} + \gamma Q_k(S_{t+1}, \arg\max_{a'} Q_{\boldsymbol{\theta}_t}(S_{t+1}, a')) - Q_{\boldsymbol{\theta}_t}(S_t, A_t)$.

---
**Algorithm 1** Deep Q-iteration (DQI)
---
Initialize weights $\boldsymbol{\theta}_0$. Initialize an empty buffer.
**for** $t \leftarrow 0, 1, 2, \dots$ **do**
    **If** $t \bmod T_{\text{eval}} = 0$ **then** $Q_k \leftarrow Q_{\boldsymbol{\theta}_t}$, update $\pi_k$ to be greedy w.r.t $Q_k$, $b_k$ to be $\epsilon$-greedy
    Choose $a_t \sim b_k(s_t)$, observe $(s_{t+1}, r_{t+1})$, and add the transition $(s_t, a_t, s_{t+1}, r_{t+1})$ to the buffer
    Sample a set of transitions $B_t$ from the buffer and update the weights:

$$\boldsymbol{\theta}_{t+1} \leftarrow \boldsymbol{\theta}_t + \frac{\alpha}{|B_t|} \sum_{(s,a,r,s') \in B_t} \delta(\boldsymbol{\theta}_t; s, a, r, s') \nabla_{\boldsymbol{\theta}} Q_{\boldsymbol{\theta}_t}(s, a)$$

    where *without* a target network:
$\delta(\boldsymbol{\theta}_t; s, a, r, s') = r + \gamma Q_{\boldsymbol{\theta}_t}(s', \pi_k(s'))) - Q_{\boldsymbol{\theta}_t}(s, a)$
    and *with* a target network:
$\delta(\boldsymbol{\theta}_t; s, a, r, s') = r + \gamma \max_{a'} Q_k(s', a') - Q_{\boldsymbol{\theta}_t}(s, a) = r + \gamma Q_k(s', \pi_k(s')) - Q_{\boldsymbol{\theta}_t}(s, a)$
**end for**
---

The central question in this work is how generalization in $Q_{\boldsymbol{\theta}}$ impacts behavior of Deep Q-iteration. Intuitively, updates to $Q_{\boldsymbol{\theta}}$ in some states may *interfere* with the accuracy of the values in other states. We formalize this notion in the next section, and in the following sections empirically connect the level of interference to performance.

## 3 Defining Interference for Value Estimation Algorithms

In this section, we define the interference measure that will be used when comparing Deep Q-iteration algorithms in the coming sections. Deep Q-iteration alternates between policy evaluation and policy improvement, where one cycle of policy evaluation and improvement is called an iteration. To explain this measure, we first need to define interference during the evaluation phase of an iteration. We then discuss interference at four different levels of granularity, the coarser of which we use for our experiments. We start at the lowest level to build intuition for the final definition of interference.

Within each iteration—in each evaluation phase—we can ask: did the agent's knowledge about its value estimates improve or degrade? The evaluation phase is more similar to a standard prediction problem, where the goal is simply to improve the estimates of the action-values towards a clear target. In the case of Deep Q-iteration with target networks, it attempts to minimize the distance to the target function $\mathbb{E}[R + \max_{a'} Q_k(S', a') | S = s, A = a]$. More generally, Deep Q-iteration, with or without target networks, attempts to reduce the squared expected TD-error: $\mathbb{E}[\delta(\theta) | S = s, A = a]^2$. Without target networks, the expected TD error is the Bellman error: $\mathbb{E}[\delta(\theta) | S = s, A = a] = \mathcal{T}^\pi Q_\theta(s, a) - Q_\theta(s, a)$, where $\mathcal{T}^\pi Q(s, a) = \mathbb{E}_\pi[R + \gamma Q(S', A') | S = s, A = a]$. A natural criterion for whether value estimates improved is to estimate if the expected TD error decreased after an update.

Arguably, the actual goal for policy evaluation within an iteration is to get closer to the true $Q^\pi(s, a)$. Reducing the expected TD error is a surrogate for this goal. We could instead consider interference by looking at if an update made our estimate closer or further from $Q^\pi(s, a)$. But, we opt to use expected TD errors, because we are evaluating if the agent improved its estimates under its own objective—did its update interfere with its own goal rather than an objective truth. Further, we have the additional benefit that theory shows clear connections between value error to $Q^\pi(s, a)$ and Bellman error. Bellman error provides an upper bound on the value error (Williams, 1993), and using Bellman errors is sufficient to obtain performance bounds for API (Munos, 2003; 2007; Farahmand et al., 2010).

**Accuracy Change** At the most fine-grained, we can ask if an update, going from $\boldsymbol{\theta}_t$ to $\boldsymbol{\theta}_{t+1}$, resulted in interference for a specific point $(s, a)$. The change in accuracy at $s, a$ after an update is

$$\text{Accuracy Change}((s, a), \boldsymbol{\theta}_t, \boldsymbol{\theta}_{t+1}) := \mathbb{E}[\delta(\boldsymbol{\theta}_{t+1}) | S = s, A = a]^2 - \mathbb{E}[\delta(\boldsymbol{\theta}_t) | S = s, A = a]^2$$

where if this number is negative it reflects that accuracy improved. This change resulted in interference if it is positive, and zero interference if it is negative.

**Update Interference** At a less fine-grained level, we can ask if the update generally improved our accuracy—our knowledge in our value estimates—across points.

$$\text{Update Interference}(\boldsymbol{\theta}_t, \boldsymbol{\theta}_{t+1}) := \max\left(\mathbb{E}_{(S,A)\sim d}\left[\text{Accuracy Change}((S,A), \boldsymbol{\theta}_t, \boldsymbol{\theta}_{t+1})\right], 0\right)$$

where $(s, a)$ are sampled according to distribution $d$, such as from a buffer of collected experience.

Both Accuracy Change and Update Interference are about one step. At an even higher level, we can ask how much interference we have across multiple steps, both within an iteration and across multiple iterations.

**Iteration Interference** reflects if there was significant interference in updating during the evaluation phase (an iteration). We define Iteration Interference for iteration k using expectation over Updated Interference in the iteration

$$\text{Iteration Interference}(k) := \mathbb{E}[X] \qquad \text{for } X = \text{Update Interference}(\boldsymbol{\theta}_{T,k}, \boldsymbol{\theta}_{T+1,k})$$

where $T$ is a uniformly sampled time step in the iteration $k$.

**Interference Across Iterations** reflects if an agent has many iterations with significant interference. Here, it becomes more sensible to consider upper percentiles rather than averages. Even a few iterations with significant interference could destabilize learning; an average over the steps might wash out those few significant steps. We therefore take expectations over only the top $\alpha$ percentage of values. In finance, this is typically called the expected tail loss or conditional value at risk. Previous work in RL (Chan et al., 2020) has used conditional value at risk to measure the long-term risk of RL algorithms. For iteration index $K$, which is a random variable,

$$\text{Interference Across Iterations} := \mathbb{E}[X | X \geq \text{Percentile}_{0.9}(X)] \qquad \text{for } X = \text{Iteration Interference}(K).$$

Iteration index $K$ is uniformly distributed and $\text{Percentile}_{0.9}(X)$ is the 0.9-percentile of the distribution of $X$. Other percentiles could be considered, where smaller percentiles average over more values and a percentile of 0.5 gives the median.

These definitions are quite generic, assuming only that the algorithm attempts to reduce the expected TD error (Bellman error) to estimate the action-values. Calculating Update Interference, however, requires computing an expectation over TD errors, which in many cases is intractable to calculate. To solve this issue, we need an approximation to Update Interference, which we describe in the next section.

## 4 Approximating Update Interference

The difficulty in computing the Update Interference is that it relies on computing the expected TD error. With a simulator, these can in fact be estimated. For small experiments, therefore, the exact Accuracy Change could be computed. For larger and more complex environments, the cost to estimate Accuracy Change is most likely prohibitive, and approximations are needed. In this section, we motivate the use of squared TD errors as a reasonable approximation.

The key issue is that, even though we can get an unbiased sample of the TD errors, the square of these TD errors does not correspond to the squared expected TD error (Bellman error). Instead, there is a residual term, that reflects the variance of the targets (Antos et al., 2008)

$$\mathbb{E}[\delta(\boldsymbol{\theta})^2 | S = s, A = a] = \mathbb{E}[\delta(\boldsymbol{\theta}) | S = s, A = a]^2 + \text{Var}\left[R + Q_{\boldsymbol{\theta}}(S', A') | S = a, A = a\right]$$

where the expectation is over $(R, S', A')$, for the current $(s, a)$, where $A'$ is sampled from the current policy we are evaluating. When we consider the difference in TD errors, after an update, for $(s, a)$, we get

$$\mathbb{E}[\delta(\boldsymbol{\theta}_{t+1})^2 | S = s, A = a] - \mathbb{E}[\delta(\boldsymbol{\theta}_t)^2 | S = s, A = a] = \mathbb{E}[\delta(\boldsymbol{\theta}_{t+1}) | S = s, A = a]^2 - \mathbb{E}[\delta(\boldsymbol{\theta}_t) | S = s, A = a]^2$$
$$+ \text{Var}\left[R + Q_{\boldsymbol{\theta}_{t+1}}(S', A') | S = a, A = a\right] - \text{Var}\left[R + Q_{\boldsymbol{\theta}_t}(S', A') | S = a, A = a\right].$$

For a given $(s, a)$, we would not expect the variance of the target to change significantly. When subtracting the squared TD errors, therefore, we expect these residual variance terms to nearly cancel. When further averaged across $(s, a)$, it is even more likely for this term to be negligible.

There are actually two cases where the squared TD error is an unbiased estimate of the squared expected TD error. First, if the environment is deterministic, then this variance is already zero and there is no approximation. Second, when we use target networks, the bootstrap target is actually $R + Q_k(S', A')$ for both. The difference in squared TD errors measures how much closer $Q_{\boldsymbol{\theta}_{t+1}}(s, a)$ is to the target after the update. Namely, $\delta(\boldsymbol{\theta}_{t+1}) = R + Q_k(S', A') - Q_{\boldsymbol{\theta}_{t+1}}(s, a)$. Consequently

$$
\begin{aligned}
\mathbb{E}[\delta(\boldsymbol{\theta}_{t+1})^2|S = s, A = a] - \mathbb{E}[\delta(\boldsymbol{\theta}_t)^2|S = s, A = a] &= \mathbb{E}[\delta(\boldsymbol{\theta}_{t+1})|S = s, A = a]^2 - \mathbb{E}[\delta(\boldsymbol{\theta}_t)|S = s, A = a]^2 \\
&\quad + \mathrm{Var}\,[R + Q_k(S', A')|S = a, A = a] - \mathrm{Var}\,[R + Q_k(S', A')|S = a, A = a] \\
&= \mathbb{E}[\delta(\boldsymbol{\theta}_{t+1})|S = s, A = a]^2 - \mathbb{E}[\delta(\boldsymbol{\theta}_t)|S = s, A = a]^2.
\end{aligned}
$$

It is straightforward to obtain a sample average approximation of $\mathbb{E}[\delta(\boldsymbol{\theta}_{t+1})^2|S = s, A = a] - \mathbb{E}[\delta(\boldsymbol{\theta}_t)|S = s, A = a]$. We sample $B$ transitions $(s_i, a_i, r_i, s'_i)$ from our buffer, to get samples of $\delta^2(\boldsymbol{\theta}_{t+1}, S, A, R, S') - \delta^2(\boldsymbol{\theta}_t, S, A, R, S')$. This provides the following approximation for Update Interference:

$$
\text{Update Interference}(\boldsymbol{\theta}_t, \boldsymbol{\theta}_{t+1}) \approx \max\left(\frac{1}{B}\sum_{i=1}^{B}\delta^2(\boldsymbol{\theta}_{t+1}, s_i, a_i, r_i, s'_i) - \delta^2(\boldsymbol{\theta}_t, s_i, a_i, r_i, s'_i), 0\right).
$$

The use of TD errors for interference is related to previous interference measures based on *gradient alignment*. To see why, notice if we perform an update using one transition $(s_t, a_t, r_t, s'_t)$, then the interference of that update to $(s, a, r, s')$ is $\delta^2(\boldsymbol{\theta}_{t+1}, s, a, r, s') - \delta^2(\boldsymbol{\theta}_t, s, a, r, s')$. Using a Taylor series expansion, we get the following first-order approximation assuming a small stepsize $\alpha$:

$$
\begin{aligned}
\delta^2(\boldsymbol{\theta}_{t+1}, s, a, r, s') - \delta^2(\boldsymbol{\theta}_t, s, a, r, s') &\approx \nabla_{\boldsymbol{\theta}}\delta^2(\boldsymbol{\theta}_t; s, a, r, s')^\top(\boldsymbol{\theta}_{t+1} - \boldsymbol{\theta}_t) \\
&= -\alpha\nabla_{\boldsymbol{\theta}}\delta^2(\boldsymbol{\theta}_t; s, a, r, s')^\top\nabla_{\boldsymbol{\theta}}\delta^2(\boldsymbol{\theta}_t; s_t, a_t, r_t, s'_t). \quad (1)
\end{aligned}
$$

This approximation corresponds to negative *gradient alignment*, which has been used to learn neural networks that are more robust to interference (Lopez-Paz et al., 2017; Riemer et al., 2018). The idea is to encourage gradient alignment to be positive, since having this dot product greater than zero indicates transfer between two samples. Other work used gradient cosine similarity, to measure the level of transferability between tasks (Du et al., 2018), and to measure the level of interference between objectives (Schaul et al., 2019). A somewhat similar measure was used to measure generalization in reinforcement learning (Achiam et al., 2019), using the dot product of the gradients of Q functions $\nabla_{\boldsymbol{\theta}}Q_{\boldsymbol{\theta}_t}(s_t, a_t)^\top\nabla_{\boldsymbol{\theta}}Q_{\boldsymbol{\theta}_t}(s, a)$. This measure neglects gradient direction, and so measures both positive generalization as well as interference.

Gradient alignment has a few disadvantages, as compared to using differences in the squared TD errors. First, as described above, it is actually a first order approximation of the difference, introducing further approximation. Second, it is actually more costly to measure, since it requires computing gradients and taking dot products. Computing Update Interference on a buffer of data only requires one forward pass over each transition. Gradient alignment, on the other hand, needs one forward pass and one backward pass for each transition. Finally, in our experiments we will see that optimizing for gradient alignment is not as effective for mitigating interference, as compared to the algorithms that reduced Update Interference.

## 5 Measuring Interference & Performance Degradation

Given a measure for interference, we can now ask if interference correlates with degradation in performance, and study what factors affect both interference and this degradation. We define *performance degradation* at each iteration as the difference between the best performance achieved before this iteration, and the performance after the policy improvement step. Similar definitions have been used to measure catastrophic forgetting in the multi-task supervised learning community (Serra et al., 2018; Chaudhry et al., 2019).

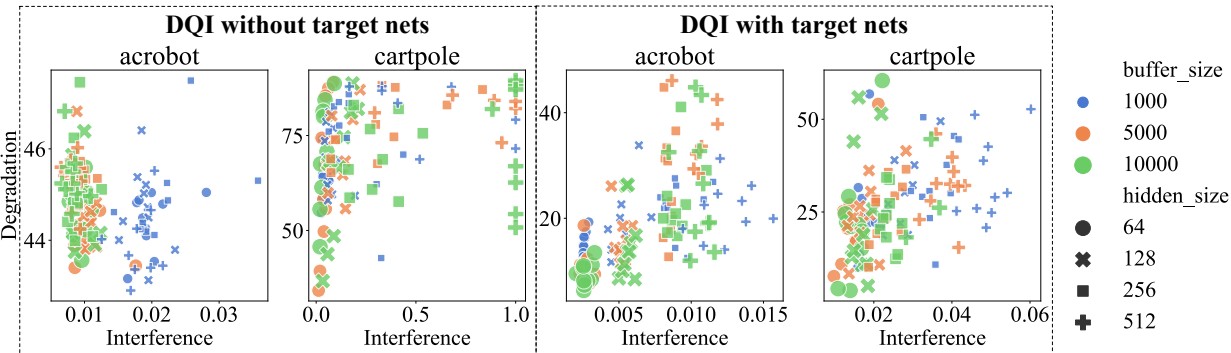

Figure 2: Correlation plot of interference and degradation with $M = 200$. Each point represents one algorithm for one run. We clip interference to be less than 1 to show all points in the plot. The results for $M \in \{100, 400\}$ are qualitatively similar to the result for $M = 200$.

Let $\mathbb{E}_{(s,a)\sim d_0}[Q^{\pi_{k+1}}(s,a)]$ be the agent performance after the policy improvement step at iteration $k$ where $d_0$ is the start-state distribution, where a random action is taken in the first step. We estimate this value using 50 rollouts. Performance Degradation due to iteration $k$ is defined as

$$\text{Iteration Degradation}(k) := \max_{i=1,\ldots,k} \mathbb{E}_{(s,a)\sim d_0}[Q^{\pi_i}(s,a)] - \mathbb{E}_{(s,a)\sim d_0}[Q^{\pi_{k+1}}(s,a)].$$

As before, we take the expected tail over all iterations. If a few iterations involve degradation, even if most do not, we should still consider degradation to be high. We therefore define Degradation across iterations as

$$\text{Degradation} := \mathbb{E}[X | X \geq \text{Percentile}_{0.9}(X)] \qquad \text{for } X = \text{Iteration Degradation}(K).$$

It might seem like Degradation could be an alternative measure of Interference. A central thesis in this paper, however, is that Interference is about estimated quantities, like values, that represent the agent's knowledge of the world. The policy itself—and so the performance—may not immediately change even with inaccuracies introduced into the value estimates. Further, in some cases the agent may even choose to forgo reward, to explore and learn more; the performance may temporarily be worse, even in the absence of interference in the agent's value estimates.

We empirically show that Interference Across Iterations is correlated with Degradation, by measuring these two quantities for a variety of agents with different buffer sizes and number of hidden nodes. We perform the experiment in two classic environments: Cartpole and Acrobot. In Cartpole, the agent tries to keep a pole balanced, with a positive reward per step. We chose Cartpole because RL agents have been shown to exhibit catastrophic forgetting in this environment (Goodrich, 2015). In Acrobot, the agent has to swing up from a resting position to reach the goal, receiving negative reward per step until termination. We chose Acrobot because it exhibits different learning dynamics than Cartpole: instead of starting from a good location, it has to explore to reach the goal.

We ran several agents to induce a variety of different learning behaviors. We generated many agents by varying buffer size $\in \{1000, 5000, 10000\}$, number of steps in one iteration $M \in \{100, 200, 400\}$, hidden layer size $\in \{64, 128, 256, 512\}$ with two hidden layers. Each algorithm performed 400 iterations. Interference Across Iterations and Degradation are computed over the last 200 iterations. A buffer for measuring Interference is obtained using reservoir sampling from a larger batch of data, to provide a reasonably diverse set of transitions. Each hyperparameter combination is run 10 times, resulting in 360 evaluated agents for Deep Q-iteration without target networks and 360 with target networks.

We show the correlation plot between Interference and Degradation in Figure 2. For DQI *with* target networks, there is a strong correlation between our measure of interference and performance degradation. For DQI *without* target networks, we actually found that the agents were generally unstable, with many suffering from maximal degradation. Measuring interference for algorithms that are not learning well is not particularly informative, because there is not necessarily any knowledge to interfere with.

We note a few clear outcomes. (1) Neural networks with a larger hidden layer size tend to have higher interference and degradation. (2) DQI with target networks has lower magnitude Interference and less degradation than DQI without target networks on both environments. Target networks are used in most deep RL algorithms to improve training stability, and the result demonstrates that using target network can indeed reduce interference and improve stability. This result is unsurprising, though never explicitly verified to the best of our knowledge. It also serves as a sanity check on the approach, and supports the use of this measure for investigation in the role of other algorithm properties that might impact interference.

## 6 Mitigating Interference via Online-aware Meta Learning

With the interference measures developed and a better understanding of some of the factors that effect interference, we now consider how to mitigate interference. In this section, we outline and empirically investigate a class of algorithms, which we call *online-aware* algorithms, that are designed to mitigate interference.

### 6.1 Online-aware Algorithms

We first discuss an objective to learn a neural network that explicitly mitigates interference. We then outline a class of algorithms that optimize this objective.

Let $\boldsymbol{\theta}$ be the network parameters and $U_B^n(\boldsymbol{\theta})$ be an inner update operator that updates $\boldsymbol{\theta}$ using the set of transitions in $B$, $n$ times. For example, $U_B^n(\boldsymbol{\theta})$ could consist of sampling mini-batches $B_i$ from $B$ for each of the $i = 1, \ldots, n$ DQI updates. The goal of online-aware learning is to update the network parameters to minimize interference for multiple steps in the future: find a direction $g_t$ at time step $t$ to minimize the $n$-step ahead Update Interference

$$\mathbb{E}_B\left[\sum_{i=1}^{|B|} \delta_i(U_B^n(\boldsymbol{\theta}_t - g_t))^2 - \delta_i(\boldsymbol{\theta}_t)^2\right]$$

Formally, we can describe the online-aware objective as

$$J(\boldsymbol{\theta}) = \mathbb{E}_B[L_B(U_B^n(\boldsymbol{\theta}))] \ \text{ where } L_B(\boldsymbol{\theta}) = \frac{1}{|B|}\sum_{i=1}^{|B|} \delta_i(\boldsymbol{\theta})^2.$$

We refer to the class of algorithms which optimizes the online-aware objective as online-aware algorithms, and provide the pseudocode in Algorithm 2. Note that this objective not only minimizes interference but also maximizes transfer (promotes positive rather than negative generalization).

The objective can be optimized by meta-learning algorithms including MAML (Finn et al., 2017), which is a second-order method by computing gradients through the inner update gradients to update the meta-parameters, or a first-order method such as Reptile (Nichol et al., 2018). Reptile is more computationally efficient since it does not involve computing higher order terms, and only needs to perform the $n$ inner updates to then perform the simple meta update.

Algorithm 2 is not a new algorithm, but rather is representative of a general class of algorithms which explicitly mitigates interference. It incorporates several existing meta learning algorithms. The choice of meta-parameters, inner update operator and meta update rules results in many variants of this Online Aware algorithm. Two most related approaches, OML (Javed & White, 2019) and MER (Riemer et al., 2018), can be viewed as instances of such an algorithm. OML was proposed as an offline supervised learning algorithm, but the strategy can be seen as instance of online-aware learning where the inner update operator updates only the last few layers at each step on a correlated sequence of data, whereas the initial layers are treated as meta-parameters and updated using the second-order method proposed in MAML.

MER, on the other hand, uses the first-order method proposed in Reptile to update the entire network as the meta-parameters. During the inner loop, MER updates the entire network with stochastic samples. MER

---

**Algorithm 2** Deep Q-iteration with Online-aware Meta Learning

---

Initialize an empty buffer. Initialize weights $\boldsymbol{\theta}_0$.
**for** $t \leftarrow 0, 1, 2, \dots$ **do**
    **If** $t \bmod T_{\text{eval}} = 0$ **then** $Q_k \leftarrow Q_{\boldsymbol{\theta}_t}$, update $\pi_k$ to be greedy w.r.t $Q_k$, $b_k$ to be $\epsilon$-greedy
    Choose $a_t \sim b_k(s_t)$, observe $(s_{t+1}, r_{t+1})$, and add the transition to the buffer $B$
    $\boldsymbol{\theta}_{t,0} \leftarrow \boldsymbol{\theta}_t$
    **for** $i \leftarrow 1, 2, \dots n$ **do**
        Sample a set of transitions $B_{i-1}$ from the buffer
        $\boldsymbol{\theta}_{t,i} \leftarrow U_{B_i}(\boldsymbol{\theta}_{t,i-1})$   # *Inner update*
    **end for**
    Meta update by second-order MAML method:
        Sample a set of transitions $B$ from the buffer
        $\boldsymbol{\theta}_{t+1} = \boldsymbol{\theta}_t + \frac{\alpha}{|B|} \sum_j \delta_j(\boldsymbol{\theta}_{t,n}) \nabla_{\boldsymbol{\theta}_t} Q_{\boldsymbol{\theta}_{t,n}}(s_j, a_j)$
    or first-order Reptile method:
        $\boldsymbol{\theta}_{t+1} = \boldsymbol{\theta}_t + \alpha(\boldsymbol{\theta}_{t,n} - \boldsymbol{\theta}_t)$
**end for**

---

introduces within-batch and across-batch meta updates; this difference to the Online-aware framework is largely only about smoothing updates to the meta-parameters. In fact, if the stepsize for the across-batch meta update is set to one, then the approach corresponds to our algorithm with multiple meta updates per step. For a stepsize less than one, the across-batch meta update averages past meta-parameters. MER also uses other deep RL techniques such as prioritizing current samples and reservoir sampling.

## 6.2 Experimental Setup

We aim to empirically answer the question: do these online-aware algorithms mitigate interference and performance degradation? We focus on an instance of the online-aware algorithm where the meta updates are performed with the first-order Reptile methods. This instance can be viewed as a variant of MER using one big batch (Riemer et al., 2018, Algorithm 6). We have also tried online-aware algorithm using MAML or only meta learning a subset of network parameters similar to Javed & White (2019), but we found online-aware algorithm using Reptile outperforms the MAML and OML variants consistently across the environments we tested.

To answer the question we compare baseline algorithms to online-aware (OA) algorithms where the baseline algorithm is DQI with or without target nets. OA treats the entire network as the meta-parameter and uses the first-order Reptile method, shown in Algorithm 2. The inner update operator uses randomly sampled mini-batches to compute the update in Algorithm 1. To fairly compare algorithms, we restrict all algorithms to perform only one update to the network parameters per step, and all algorithms use similar amounts of data to compute the update. We also include two other baselines: *Large* which is DQI with 10 to 40 times larger batch sizes so that the agent sees more samples per step, and *GA* which directly maximizes gradient alignment from Equation equation 1 within DQI[3].

## 6.3 Experiments for DQI without Target Networks

We first consider DQI without target networks, which we found in the previous section suffered from more interference than DQI with target networks. We should expect using an online aware update should have the biggest impact in this setting. Figure 3 summarizes the results on Acrobot and Cartpole.

We can see that OA significantly mitigates interference and performance degradation, and improves control performance. Large (light-blue) and GA (green) do not mitigate interference nearly as well. In fact, Large generally performs quite poorly and in two cases actually increases interference. Our results indicate that

---

[3]In fact, both MAML and Reptile approximately maximize the inner product between gradients of different mini-batches (Nichol et al., 2018).

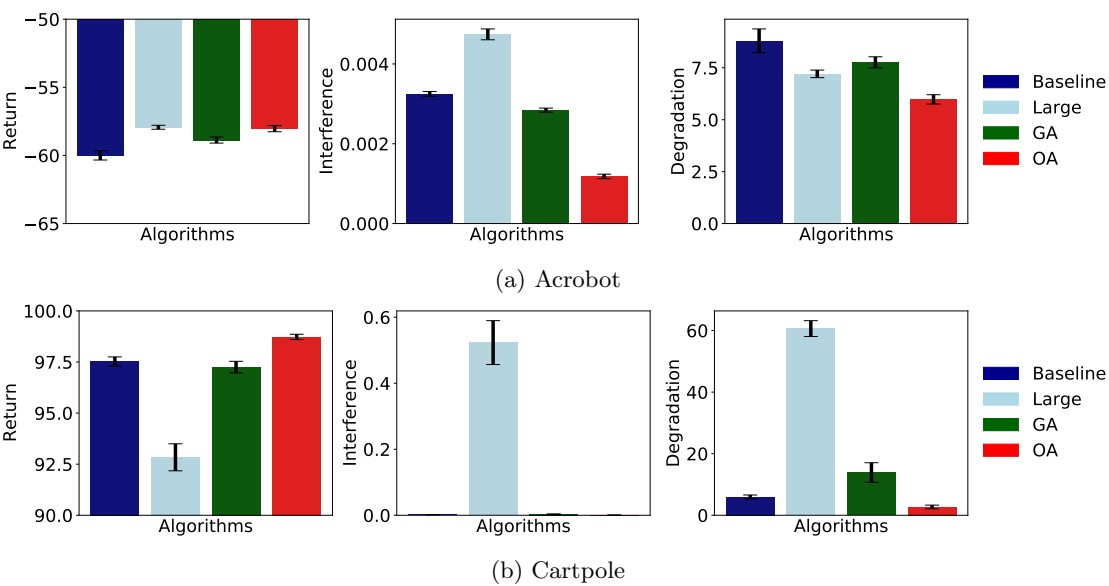

Figure 3: Acrobot and Cartpole results for DQI without target nets. Return, Interference and Degradation are computed for the last 200 iterations, and averaged over 30 runs with one standard error.

the online-aware algorithms are capable of mitigating interference, whereas, simply processing more data or directly maximizing gradient alignment are not sufficient to mitigate interference.

Further insight can be found by investigating data from individual runs. The previous results aggregate performance and return over runs, which can remove much of the interesting structure in the data. Looking closer, Figure 4(a) shows the return per run (left) and iteration interference per run (right) in Acrobot, revealing that vanilla DQI without target nets (in blue) experienced considerable problems learning and maintaining stable performance. OA (in red) in comparison was substantially more stable and reached higher performance. Overall OA also exhibits far less interference.

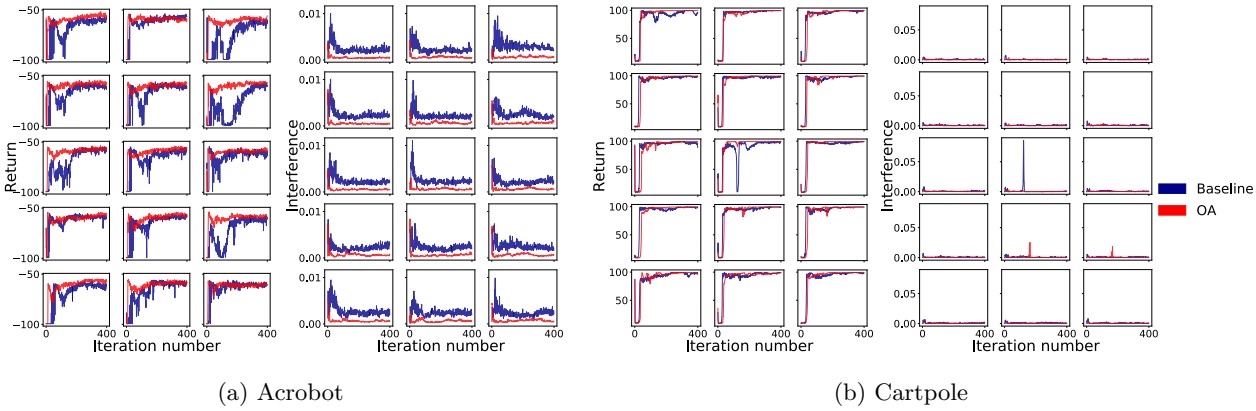

Figure 4: Learning curves and Iteration Interference, one run per subplot, for DQI *without* target networks (blue) compared to adding the online-aware objective (red).

## 6.4 Experiments for DQI with Target Networks

In this section, we investigate the utility of OA for DQI with target networks. Again, it is unlikely to be particularly useful to add OA for settings where the interference is low. In the previous section, in Figure 2, we found that DQI with target networks had higher interference with a larger hidden layer size (512). We

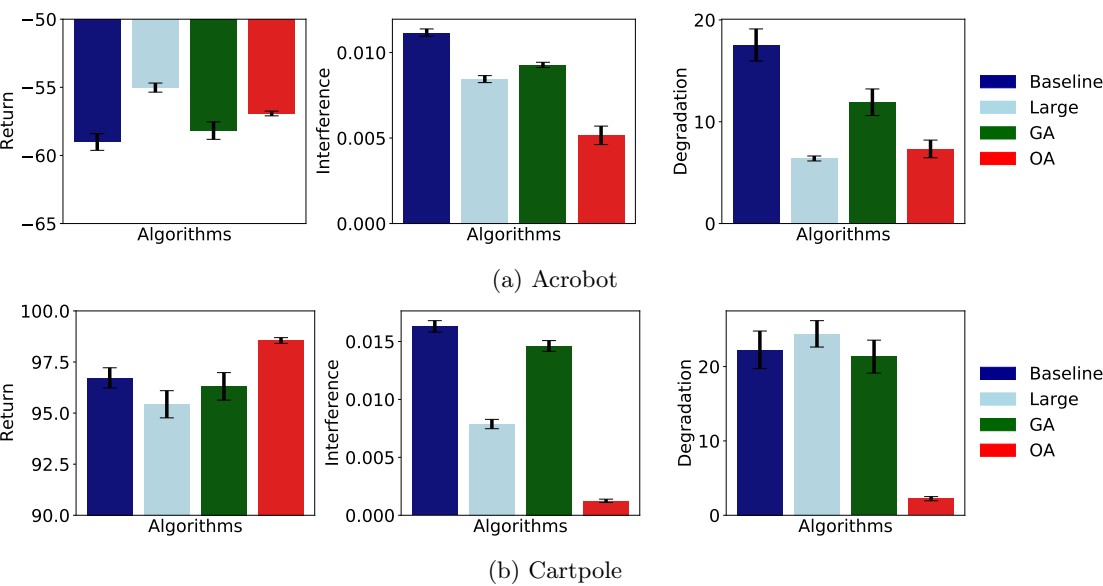

Figure 5: Acrobot and Cartpole results for DQI with target nets. Return, Interference and Degradation are computed for the last 200 iterations, and averaged over 30 runs with one standard error.

therefore test the benefits of OA for this larger network size in this experiment. Figure 5 summarizes the results on Acrobot and Cartpole.

We can see that the addition of OA to DQI with target networks helps notably in Cartpole, and only slightly in Acrobot. This is in stark contrast to the last section, where there was a large gain in Acrobot when adding OA. This outcome makes sense, as when adding OA to DQI *without* target networks, the agent went from failure to learning reasonably well. In this case of DQI *with* target networks, the agent was already learning reasonably. Nonetheless, the addition of the OA objective does still provide improvement. In Cartpole, the improvement is more substantial. Again, looking at the previous correlation plots in Figure 2, we can see that a hidden layer size of 512 resulted in more interference in Cartpole, and more Performance Degradation; there was more room for OA to be beneficial in Cartpole. When looking at the individual runs in Figure 5, we can see that DQI has some drops in performance, whereas the OA variant is much more stable.

A few other outcomes are notable. The larger network (10x the size) was actuthally better than OA in Acrobot but did worse than the base algorithm (DQI with hidden layer sizes of 512) in Cartpole. The most consistent performance was with OA. Further, except for the larger network, there was a clear correspondence between interference and performance: OA reduced interference most and performed the best, GA was next in terms of both and then finally the baseline with no additions.

## 7   Conclusion

In this paper, we proposed a definition of interference for value-based methods that fix the target policy for each iteration. We justified the use of squared TD errors to approximate this interference and showed this interference measure is correlated with control performance. In this empirical study across agents, we found that target networks can significantly reduce interference, and that bigger hidden layers resulted in higher interference in our environments. Lastly, we discuss a framework for online-aware learning for Deep Q-iteration, where a neural network is explicitly trained to mitigate interference. We concluded with experiments on classical reinforcement learning environments that showed the efficacy of online-aware algorithms in improving stability and lowering our measure of interference. This was particularly the case for Deep Q-iteration without target networks, where interference was the highest. These online aware algorithms also exhibit lower performance degradation across most of the tested environments.

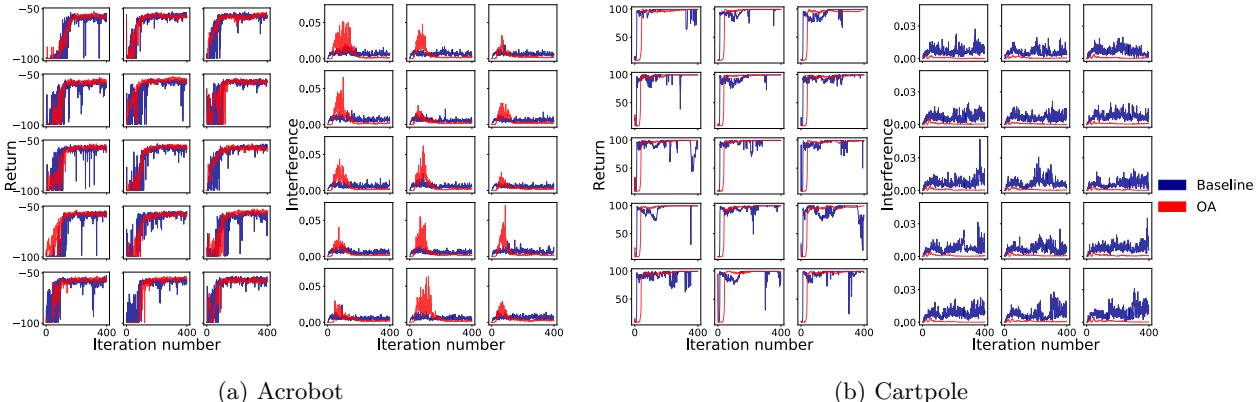

(a) Acrobot  (b) Cartpole

Figure 6: Learning curves and Iteration Interference, one run per subplot, for DQI *with* target networks (blue) compared to adding the online-aware objective (red).

There are several limitations in this work. We did not carefully control for other factors that could impact performance, like exploration or the distribution of data in the replay buffer. DQI without target networks performed poorly in Acrobot under many hyperparameter settings, making it difficult to measure interference. Later, by including online-aware learning, the performance significantly improved, suggesting interference was indeed the culprit. But, it was difficult to perfectly identify, at least using only our measure. The correlation plots themselves indicate there are other factors, beyond interference, driving performance degradation.

Another important limitation is that we only examined Deep Q-iteration algorithms, which fixed the behavior during each iteration. Allowing this behavior to update on each step, to be $\epsilon$-greedy with respect to the current action-values, would give us DQN. An important next step is to analyze this algorithm, and other extensions of Deep Q-iteration.

Finally, these results highlight several promising avenues for improving stability in RL. One surprising outcome was the instability, within a run, of a standard method like Deep Q-iteration. The learning curve was quite standard, and without examining individual runs, this instability would not be obvious. This motivates re-examining many reinforcement learning algorithms based on alternative measures, like degradation and other measures of stability. It also highlights that there are exciting opportunities to significantly improve reinforcement learning algorithms by leveraging online-aware learning.

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

# A   Experimental details

## A.1   Experiment setup

We experiment with two environments: Cartpole and Acrobot from the OpenAI gym (`https://gym.openai.com/`). We set the maximum steps per episode to 500, and use a discounting factor $\gamma = 0.99$ in all environments.

We use 50 Monte Carlo rollouts to estimate the performance of the policy at each iteration, that is, $\mathbb{E}_{(s,a) \sim d_0}[Q^{\pi_k}(s,a)]$. For evaluating the TD error difference, we use a reservoir buffer of size 1000, which approximates uniform sampling from all the past transitions.

## A.2   Network architecture and hyperparameters

For all experiments, we use a three-layer neural network with ReLU activation (Glorot et al., 2011) and He initialization (He et al., 2015) to initialize the neural. For Adam and RMSprop optimizer, we use the default values for the hyper-parameters except the step size.

For the experiments in Section 5, we generate a set of hyper-parameter by choosing each parameter in the set:

- Batch size $= 64$

- Step size $\alpha = 0.0003$

- Number of iteration $= 400$

- Optimizer $=$ Adam

- Buffer size $\in \{1000, 5000, 10000\}$

- Hidden size $\in \{64, 128, 256, 512\}$

- Number of steps in an iteration $M \in \{100, 200, 400\}$

- Network architecture $\in \{\text{action-output}, \text{action-input}\}$

For the experiments in Section 6.3 and 6.4, all algorithms use buffer size of 10000, 100 steps in an iteration, and 400 iterations. DQI without target nets uses hidden size of 128, and DQI with target nets uses hidden size of 512. The best parameters are chosen based on average performance of the policies over the last 200 iterations.

**Baseline.**   We sweep the hyperparameters for DQI in the range:

- Batch size $= 64$

- Optimizer $\in \{\text{Adam, RMSprop}\}$

- Step size $\alpha \in \{0.003, 0.001, 0.0006, 0003, 0.0001, 0.00001\}$

**DQI with large batch size.**   For the baseline Large, we find the best batch size in the range:

- Batch size $\in \{640, 1280, 2560\}$

**Online-aware DQI.** In our experiment, We sweep over the hyperparameters in the set:

- Inner update optimizer = SGD

- $\alpha \in \{1.0, 0.3, 0.1, 0.03\}$

- $\alpha_{\text{inner}} \in \{0.01, 0.001, 0.0001, 0.00001\}$

- Number of inner updates $K \in \{5, 10, 20\}$

**DQI maximizing gradient alignment (GA).** When updating the parameters, we draw two mini-batch samples $B_1$ and $B_2$ and add a regularization term in the loss function:

$$-\lambda \left[ \frac{1}{|B_1|} \sum_{i \in B_1} \nabla_{\boldsymbol{\theta}} \delta_i^2(\boldsymbol{\theta}) \right]^\top \left[ \frac{1}{|B_2|} \sum_{j \in B_2} \nabla_{\boldsymbol{\theta}} \delta_j^2(\boldsymbol{\theta}) \right],$$

normalized by the number of parameters in the network. In our experiment, We sweep over the hyperparameters in the set:

- Optimizer $\in \{\text{Adam, RMSprop}\}$

- Step size $\alpha \in \{0.003, 0.001, 0.0006, 0003, 0.0001, 0.00001\}$

- $\lambda \in \{10.0, 1.0, 0.1, 0.01, 0.001\}$

