# OpenReview forum: "Measuring and mitigating interference in reinforcement learning"
_TMLR — Rejected by TMLR_

### Review · Reviewer_ibtD · 2022-08-16

**Summary Of Contributions:**

The aim of the paper is to better understand catastrophic interference in the context of value-based RL. To do so, the authors introduce a novel measure of interference and show that it correlates with a proposed measure of 'degradation', which measures instability in control performance.
Secondly, they describe a class of learning algorithms which they call "online-aware" algorithms. These algorithms are strongly related to a class of meta learning algorithms, but motivate them from the perspective of interference reduction. They show that these methods improve performance and reduce the introduced measure of interference.

**Broader Impact Concerns:**

N/A.

**Requested Changes:**

* Please reply to my "Main Point" above about whether the current way of how Interference is measured makes sense and whether the correlation measurements with "Degradation" make sense.
* If there is any merit to this "Main Point", I believe it would be useful to include a corresponding section into the paper. I'm not sure what exactly this should discuss, my initial idea would be: How does "Interference" relate to the distribution over observed accuracy changes and how does the proposed Interference measure measure this (or not) and what other effects could influence the observed measure.
* If possible, please include additional experiments on higher-dimensional environments (for example Ant, Humanoid or Atari) to verify the results on more complex problems.

**Strengths And Weaknesses:**

## Strengths

* This is a very interesting topic that is highly relevant to the foundations of RL. As such, any improvement in understanding would have great impact
* The writing is good. Everything is explained clearly. I also liked that the related work is "interwoven" at suitable locations.
* I liked the discussion about possible biases in section 4.


## Weaknesses

### Small points

As cliche as this request is, I do believe that experiments on more domains would strenghten the paper. Especially since the paper is trying to make general statements about interference in RL, it would be good to support their claims on either higher-dimension control problems (for example Ant/Humanoid) or visual environment (e.g. Atari). (Currently experiments are only performed on acrobot and cartpole).

Related work: I think you've missed the "deadly triad" literature which seems relevant to me, both the original Sutton & Barto one and Van Hasselt, Hado, et al. "Deep reinforcement learning and the deadly triad."

### Main point

However, my main point is that I am unsure about how useful the definition for interference is that is being used. I'm in no way certain about my assessment here. I will try to make my reasoning as clear as possible and am looking forward to the authors' response.

To recap, the proposed interference is

$$
\text { Interference Across Iterations }:=\mathbb{E}\left[X \mid X \geq \text { Percentile }_{0.9}(X)\right]
$$

where $X$ is the average *positive* change in squared TD errors between updates. Negative changes (i.e. measured improvements in the TD error over updates) are ignored.

Furthermore, as I understand it, this is not measured at the end of training once training has reached a steady-state, but using a buffer which contains samples collected uniformly across training (A.1: "For evaluating the TD error difference, we use a reservoir buffer of size 1000, which approximates uniform sampling from all the past transitions.").

If we think about the distribution of observed Accuracy Changes, then a higher Interference value can be caused by either:

1. Higher Variance
2. Higher mean of the distribution

and reversely, reduction in Interference could be caused by a reduced variance or lower mean. A lower mean here means basically an algorithm that learns faster.

The variance is probably roughly what we want to measure, i.e. high variance probably indicates high instability of training updates, i.e. probably high interference. However, this line of reasoning could be made clearer. Furthermore, there might be confounding factors: For example, the variance could be partially influenced by how we measure it (number of samples or how they are drawn).

In addition to an imprecise connection between "Interference" and "Var(Accuracy Change)", I believe the dependence on the mean could be a serious problem: After all, meta-learning algorithms are optimised for faster learning, so the lower measured Interference could just be due to faster learning (hence a lower average change in accuracy, hence a lower measured Interference) instead of a lower variance of the accuracy change.

Lastly, I wonder how meaningful the correlation with 'Degradation' is. Since Degradation is again measured using the 0.9 percentile, it should be also influenced by the variance of the training, so maybe we're just measuring the same thing in two different ways?

As I said, this is my current understanding, but I'm looking forward to hear the authors' response.

---

> ### Author Response · Authors · 2022-08-23
> **Response to Reviewer ibtD**
>
> Thank you for the reviews and comments. We would like to first answer your questions on the definition of interference:
>
> >this is not measured at the end of training once training has reached a steady-state, but using a buffer which contains samples collected uniformly across training
>
> We want to first clarify that there are measures of interference at different granularities (for each data point, for each update, for each iteration, and across iteration). In our experiment, we measure Interference Across Iteration for the last 200 iterations after the agent has reached a certain performance. The Interference is computed on a dataset that covers the state-action space well to reflect the agent’s own knowledge about the world.
>
> >The variance is probably roughly what we want to measure, i.e. high variance probably indicates high instability of training updates
>
> Accuracy Change is defined for a specific point $(s,a)$, and we found the mean of Accuracy Change better reflects Interference for a single update. Based on our own experiment, we found that for a single update, the distribution of Accuracy Change is likely to be unimodal with mean close to zero or bimodal with a positive or negative mean. That is, each update would result in either of (1) do not change Accuracy Change for most points (no interference) (2) increase Accuracy Change for some points and make no effect on other states (interference) (3) decrease Accuracy Change for some points and make no effect on other states (generalization). As a result, taking the average over all data points better summarizes whether each update causes interference or not. On the other hand, a high variance might not necessarily imply that the agent is suffering from interference. For example, the distribution of Accuracy Change might be bimodal with some points near zero and some points are large negative. In such a case, the variance of Accuracy Change is large but the agent actually has fast learning instead of interference.
>
> >After all, meta-learning algorithms are optimised for faster learning, so the lower measured Interference could just be due to faster learning (hence a lower average change in accuracy, hence a lower measured Interference) instead of a lower variance of the accuracy change.
>
> We want to clarify that meta-learning methods both minimize multi-step Update Interference and maximize transfer (positive generalization).
>
> >I wonder how meaningful the correlation with 'Degradation' is. Since Degradation is again measured using the 0.9 percentile, it should be also influenced by the variance of the training, so maybe we're just measuring the same thing in two different ways?
>
> As we described in the paper, it might seem like Degradation could be an alternative measure of Interference. However, Interference is about estimated quantities, like values, that represent the agent’s knowledge of the world. Degradation on the other hand cannot be measured by the agent unless the agent has access to a simulator—e.g., it is not readily measurable on a robot or in industrial control settings. The policy itself may not immediately change even with inaccuracies introduced into the value estimates. Further, in some cases, the performance may temporarily be worse, even in the absence of interference in the agent’s value estimates.
>
> >Especially since the paper is trying to make general statements about interference in RL, it would be good to support their claims on either higher-dimension control problems (for example Ant/Humanoid) or visual environment (e.g. Atari).
>
> We introduce a new methodology to measure interference, to help investigate algorithms across many environments. The critical components of the paper are to run clear experiments which can be built on, and in some cases to introduce re-usable methodologies. To use experiments to show generality, we need to test it on a significant number of environments.  We cannot conclusively state anything about the behavior of these agents across the many different problems an RL agent could see. This set is vast. Instead, we provide sound evidence in two classic but diverse environments, and a repeatable methodology for others to continue the work.
>
> **Please let us know if our responses do not fully answer your questions or if there are other changes we should make! Thank you!**

---

> ### Author Response · Authors · 2022-09-27
> **Follow-up response**
>
> We would like to better understand your perspective. To summarize, we believe you are concerned that the variance of accuracy change is actually more reflective of interference, but we are taking the mean of the upper percentile. Further, you state that faster learning might make interference look smaller, even if the variance in accuracy change is higher.
>
> To clarify: we measure interference after an initial learning phase. Therefore our results are not skewed by how quickly the algorithm learns in the beginning.
> Additionally, we believe you may misunderstand how the buffer is used as well as how Interference Across Iterations is computed. Update Interference is a regular average of Accuracy Change (diff in td errors) across the buffer (both positive and negative). The Interference Across Iterations measures this Update Interference during learning, and takes the average across iterations (time) of the top percentile of max(UpdateInterference_t(buffer), 0). A lower mean can indicate that the algorithm learned slowly but had very little interference (very little positive Update Interference so many zero when considering this upper percentile).

---

> > ### Comment · Reviewer_ibtD · 2022-10-02
> > **Thank you for your response**
> >
> > Thank you for replying to my concerns. I would like to clarify that my main concern is whether the proposed measure of interference measures what we want it to.
> >
> > In particular, my concerns are:
> >
> > 1. The proposed measure of interference, when measured while the agent is still improving, depends also on the rate of improvement of the agent as this shifts the mean of the distribution of Accuracy Change. You replied that interferences is measured "for the last 200 iterations after the agent has reached a certain performance. ", but that doesn't sound like you're making sure the agent has already reached it's final performance. You also mention in your reply that the distribution of AC depends on the learning speed. However, I don't believe a measure of interference should depend on the learning speed of the agent.
> >
> > 2. Your choice of taking the 0.9 percentile is not very well motivated and can induce weird effects into how the interference is measured. For example, Interference Across Iterations now depends on the size of the dataset *d* that is used to compute the Update Interference: If d is smaller, that means the variance of the estimated Accuracy Change is larger, which makes the expected Interference Across Iterations also larger (because you're only looking at one tail of the distribution)!
> >
> > Currently I'm not convinced the measure, as proposed, makes sense.

---

> > > ### Author Response · Authors · 2022-10-05
> > > **Response to Reviewer ibtD**
> > >
> > > We would like to address the primary issue here, as it seems that there might be some misunderstanding about how we compute the measure.
> > > 1. We obtain a relatively large test dataset (10000 samples) of tuples (s,a,s’,r), using reservoir sampling, to reasonably cover the space.
> > > 2. The Update Interference is computed by taking the difference in average TD errors across the dataset. (Note Accuracy Change is the change for one of the tuples in the dataset, and it is possible you are using the term Accuracy Change for what we are calling Update Interference).
> > > We use a large dataset (10000 samples) to estimate this expectation for Update Interference. We believe you are saying that Update Interference is an estimator, since it uses only a sample of data rather than the true Update Interference across all possible tuples, and this sample average estimator could be a high-variance estimator if our test dataset is small. But 10000 samples for these problems is quite large, and likely a low-variance estimator of the true Update Interference.
> > > 3. We get a good estimate of Update Interference after every iteration. In our setting we have 200 points, corresponding to the last 200 iterations. We take the 0.9 percentile across these 200 points because it allows us to see: was there substantial interference in at least 10% of the iterations? If we simply averaged these points, for example, 180 iterations may have very little interference but the behavior of the agent is dominated by the significant interference on the other 20 iterations. The average would not at all capture that, and would indicate low interference. The 0.9 percentile, on the other hand, would reflect this.
> > >
> > > Does this clarify? We did not precisely understand the issue with the tail of the distribution that you mentioned, so if you could further clarify, we would be happy to respond.
> > >
> > > > that doesn’t sound like you’re making sure the agent has already reached its final performance
> > >
> > > The reason we used 200 iterations is that in these classic environments the agent can achieve a good performance within 200 iterations. We agree that for other environments we might need more than 200 iterations. We will clarify this in the paper.
> > >
> > > > You also mention in your reply that the distribution of AC depends on the learning speed. However, I don’t believe a measure of interference should depend on the learning speed of the agent.
> > >
> > > The distribution of AC depends on the learning rate of each update if that is what the reviewer is referring to. If we can set the learning rate to zero once the agent has learned an optimal policy, then the measure of interference should be zero. We would appreciate it if the reviewer can clarify why the measure should not depend on the learning rate.
> > >
> > > On the other hand, the interference measure does not depend on whether the agent is learning fast or slow. An agent can have fast learning and high interference at the same time, or slow learning with high interference. What our measure is capturing is whether each update causes a decrease in accuracy.  Ideally, no matter whether the agent is learning fast or slow, we  do not want each update to cause a decrease in accuracy.

---

> > > > ### Comment · Reviewer_ibtD · 2022-10-06
> > > > **Thank you**
> > > >
> > > > Thank you for your reply!
> > > >
> > > > I've drawn my understanding of what's happening: https://ibb.co/whmtNV1
> > > >
> > > > In **black** i've drawn one case:
> > > > 1. Top left image: There's a random distribution over AC (top left): P(AC)
> > > > 2. Top middle image: The expectation $E_d[AC]$ is still a random variable, drawn from P(E[AC]), with randomness coming from the current theta, the current iteration, what exactly is in the dataset d, etc..
> > > > 3. Top right image: To get P(UI), we map all negative values of E[AC] to 0.
> > > > 4. Bottom left: To get P(II), we average over T. I've highlighted the 0.9 percentile in red.
> > > > 5. Bottom middle: Distribution of P(IAI), i.e. the average value of the percentile.
> > > >
> > > > In **green** I've drawn what I believe would happen if the distribution P(AC) would have higher variance, which would ultimately lead to higher average P(IAI). Something similar would happen if the distribution P(AC) would be shifted to the right.
> > > >
> > > > So bottom line is: The measured IAI depends on how the distribution P(AC) looks like. In turn, P(AC) depends on a lot of factors, including the learning rate (influencing variance), but also the size of $d$ or the mean of P(AC), i.e. whether the agent is still improving or whether it has reached its steady state equilibrium.
> > > >
> > > > So my worry is, that IAI is not a reliable measurement because changes in IAI could be caused by things *other* than what we want to measure.

---

> > > > > ### Author Response · Authors · 2022-10-07
> > > > > **Response to Reviewer ibtD**
> > > > >
> > > > > We very much appreciate this response!
> > > > >
> > > > > It is a great idea to be more clear about sources of variability. We are estimating E[IAI], the expected Interference Across Iteration, because we average the IAI across runs (30 runs). The sources of variability for our estimator are:
> > > > > 1. The number of runs for averaging IAI, for a given agent (with all hyperparameters fully specified).
> > > > > 2. Within a run (seed), we have one agent that runs for T iterations. Conditioning on the seed, we can generate the corresponding distribution over T Iteration Interference (for this seed and agent). Though we do not exactly sample Iteration Interference (it is deterministically obtained because the seed is fixed), we can still reason about the resulting density over Iteration Interference and get a percentile to compute IAI.
> > > > > 3. Within an iteration, we have the set of (theta, theta’) pairs for which we compute the Update Interference. Iteration Interference is simply the average of the UI over the set of (theta, theta’) pairs.
> > > > > 4. The source of variability in the Update Interference, conditioned on seed and (theta, theta’) pair, is only due to the variability in estimating the expected Accuracy Change from a finite dataest. As a sample average, we know that the variance of E[AC] is of order O(1/n) where n is the number of samples in the buffer. The variance of AC would impact the final interference measure, but the impact is negligible.
> > > > >
> > > > > Your picture, as drawn, suggests we are reasoning about E[AC] *across all pairs theta, theta’ and across all iterations* and then taking the positive quadrant for that distribution over E[AC]. But, we are taking E[AC | theta, theta’], which is a very narrow distribution. Then we take UI = max(E[AC | theta, theta’], 0), again for a given theta, theta’ pair. Finally, the distribution we consider for IAI takes all of these computed UI, and has very little source of variability for each estimate of UI for a theta, theta’ pair. The variability across pairs for UI could be broad, because naturally UI can change quite a bit depending on the pair we compute UI for. But that distribution is just reflecting the change in parameters (which is exactly what we want for interference) not reflecting errors in statistical estimation.
> > > > >
> > > > > We also want to clarify that the mean of AC is what we want to measure when computing UI. The mean of AC is an indicator of how one update impacts the data points sampled from a distribution. A higher mean of AC implies most data points have worse accuracy after on update. A negative mean of AC means most data points improve the accuracy after one update. The variance of AC is negligible as explained earlier.

---

> > > > > > ### Comment · Reviewer_ibtD · 2022-10-08
> > > > > > **Thank you**
> > > > > >
> > > > > > Thank you very much for your reply, which I agree with and helped clarify things!
> > > > > >
> > > > > > There's one part I'm still unsure about. From your earlier reply:
> > > > > >
> > > > > > > On the other hand, the interference measure does not depend on whether the agent is learning fast or slow.
> > > > > >
> > > > > > But doesn't the learning speed (of the value function) directly impact the expected AC? I.e. the faster the agent learns, the more negative the average AC should be?

---

> > > > > > > ### Author Response · Authors · 2022-10-10
> > > > > > > **Response to Reviewer ibtD**
> > > > > > >
> > > > > > > Thank you so much for your response!
> > > > > > >
> > > > > > > When we compute UI, we clip the expected AC at zero. That is, we only want to measure the “interference” part, not the “transfer” part. Also there is a distinction between learning speed (learning fast is good!) and the learning rate (stepsize). It is true that if we set the stepsize to a large value, then the agent is likely to suffer from high interference even it might learn fast.

---

> > > > > > > > ### Comment · Reviewer_ibtD · 2022-10-10
> > > > > > > > **Response**
> > > > > > > >
> > > > > > > > Thank you for your reply!
> > > > > > > >
> > > > > > > > Re clipping: But you clip the *expected* AC. So in my understanding, one update of $\theta$, will do three things:
> > > > > > > >
> > > > > > > > * Induce a negative AC on the state-action pairs it was trained on
> > > > > > > > * Induce a negative AC on state-action pairs it generalises to
> > > > > > > > * Induce a positive AC on state-action pairs it interferes with.
> > > > > > > >
> > > > > > > > By averaging over $d$ before clipping, you average all three things. Hence, if your agent is learning faster, it will lead to a smaller (i.e. more negative) AC on the states in was trained on (and maybe also the ones it generalises to). Consequently, even if it has the same amount of interference than another agent that learns slower, the faster-learning agent will have an overall smaller UI, because the smaller AC on the trained-on/generalised-to states will drag down the average AC.
> > > > > > > > And I don't think the late clipping here changes things:
> > > > > > > > * For average AC values that remain positive for the faster-learning agent, it makes no difference
> > > > > > > > * For average AC values that are negative for both agents, it makes no difference
> > > > > > > > * Only for average AC values that are positive for the slower-learning agent but negative for the faster learning agent the clipping has an effect (in this hypothetical comparison between two agents). But even then it only reduces the decrease of UI of the faster learning agent, not eliminate it completely.

---

> > > > > > > > > ### Author Response · Authors · 2022-10-11
> > > > > > > > > **Response to Reviewer ibtD**
> > > > > > > > >
> > > > > > > > > Our goal was to ask: across states, did the update improve performance? This means we average across all state-action pairs, including those similar to the ones we trained on (not the same since the buffer is a separate reservoir) and those that the agent generalizes well to. We believe you might be suggesting that we should only average the AC when it is positive, namely average max(AC, 0). This would reflect the level of interference, on that state-actions where there was interference. Of course, one could easily test this version of the measure too, using the methodology we laid out (and in fact, we did test it ourselves). However, we believe expected AC is a better reflection for the reason already mentioned: it gives an estimate of how much the update improved or harmed performance overall. Just looking at max(AC, 0), then one agent that has interference across many state-actions might actually look like it has lower interference than one that has a couple of state-action pairs with higher interference. At this point, this just might be an issue with the fact that the word interference is a general, laymans word that could have different meanings. We precisely defined it the way we have, with expected AC, for these reasons mentioned. It is possible another person might want interference to be only about the level of interference for the state-actions where interference occurs (and so uses expected max(AC, 0)). It is hard to say there is precisely one right answer for what the word interference should mean. We propose one reasonable measure here. Another work could make a case for another measure, or even a complementary measure where we report both to get more insight.

---

> > > > > > > > > > ### Comment · Reviewer_ibtD · 2022-10-12
> > > > > > > > > > **Response**
> > > > > > > > > >
> > > > > > > > > > Thank you for your reply!
> > > > > > > > > >
> > > > > > > > > > > Our goal was to ask: across states, did the update improve performance?
> > > > > > > > > >
> > > > > > > > > > But that is not how I understood how the paper defines interference. For example, from the following quotes from the paper, I understood that with *interference* you mean specifically the *negative* impact on other states, not the *average* impact:
> > > > > > > > > >
> > > > > > > > > > > This change resulted in interference if it is positive, and zero interference if it is negative.
> > > > > > > > > >
> > > > > > > > > > > Interference classically refers to an update negatively impacting the agent’s previous learning
> > > > > > > > > >
> > > > > > > > > > And this also corresponds more to my understanding of what 'to interfere' means. Note that I'm not suggesting to use max(AC, 0) instead - it just seems to me that your proposed IAI might not exactly measure what would be understood by interference after reading the paper.

---

> > > > > > > > > > > ### Author Response · Authors · 2022-10-13
> > > > > > > > > > > **Response to Reviewer ibtD**
> > > > > > > > > > >
> > > > > > > > > > > Ok, we can see how this sentence might have implied that. We will try to avoid making such implications in the work. We would like to note, though, that these two sentences were in different parts of the paper in different contexts. This one was in the introduction:
> > > > > > > > > > > > Interference classically refers to an update negatively impacting the agent’s previous learning
> > > > > > > > > > >
> > > > > > > > > > > This sentence reflects that we care about how an update impact the agent’s previous learning overall, which is the average AC for all data points the agent has seen. We do not mean that the interference has to be defined for a single data point or the data points that are negatively impacted. It is in the introduction, motivating the work, rather than in the technical section beside the definition of Update Interference.
> > > > > > > > > > >
> > > > > > > > > > > The other sentence comes right after defining AC.
> > > > > > > > > > > > This change resulted in interference if it is positive, and zero interference if it is negative.
> > > > > > > > > > >
> > > > > > > > > > > This sentence defines interference for a single data point (AC), as we need to define the notion of interference at different granularities. However, we don’t propose to directly use AC for a single data point as the final interference measure. We need to summarize interference across all data points, and we propose to use UI = max(E(AC), 0).
> > > > > > > > > > >
> > > > > > > > > > > We do see how using the term interference more generically, even at different granularities, could cause confusion. We will adjust the text to be careful about this, and for this sentence for example we might say "this changed resulted in interference at the level of one datapoint…".

---

### Review · Reviewer_FBid · 2022-08-17

**Summary Of Contributions:**

The paper proposes:
- measuring inference on the level of the TD-error
- presents an empirical analysis of how the proposed metric correlates with the performance
- proposes a 'inference-free method'

**Requested Changes:**

See the questions and comments in the previous section.

**Strengths And Weaknesses:**

Strengths:
- a well-chosen important problem
- a solid clean pipeline with analysis and a new method
- taking care of statistical relevance (30 runs)

Weaknesses - a general comment is that I find the paper unconvincing. Some of my concerns might be due to my lack of proper understanding. I'd be happy to raise my score once these have been clarified.

- How much the proposed metric is different from monitoring of the (squared) td-error (which is the principal metric for any value method)? In other words, wouldn't the conclusion be the same if we take the td-error, say for the sake of simplicity in the deterministic environments?
- I find the conclusion of Section 5 unsatisfactory. I'd say that correlation in the 'without target nets' is disputable. It does not also seem that solid for the other case. At the very least, I'd like to see a more proper analysis, perhaps over the groups.
- Following that, having only two envs in the empirical works feels risky (I acknowledge, however, a solid job of sweeping over hyperparameters).
- The differences in returns seem not that big in Fig 5 (are they?) Is the gain significant enough to justify the higher computational cost?
- Could more baselines be included (e.g., the one listed in Sec 6.1). It is fine if these are in the appendix.

Some minor comments:
- I find it valuable that the proposed method produces smoother curves (Fig 4). I'd suggest putting a little bit more attention to this fact (and perhaps measuring it more).
- I like the didactical example in the introduction. I am not sure how much we can relate this to the more serious environments studied further.
- It'd be great to relate the proposed metric to the specific violation of the i.i.d. assumptions listed in the second paragraph of the introduction.
- I find some parts slightly too long, e.g. the 5 and 6 paragraphs of the intro, Section 5 etc.
- The authors should perhaps comment more on the correlation of the Bellman error and value error (in the view of [1])
- It'd be good to have an explicit measurement of the computational overhead due to the use of the meta-learning methods.



[1] Why Should I Trust You, Bellman? The Bellman Error is a Poor Replacement for Value Error.

---

> ### Author Response · Authors · 2022-08-23
> **Response to Reviewer FBid (2/2)**
>
> >Explicit measurement of the computational overhead of the meta-learning methods
>
> We first want to clarify that we use the meta-learning methods to: (1) demonstrate the proposed measure is meaningful, and (2) demonstrate that interference is indeed harming performance in practice and there are effective approaches to mitigate interference.
>
> We measured the computation time to complete 400 iterations for all methods on Acrobot (based on our Pytorch implementation and a machine with only CPUs). The meta-learning with 20 inner updates takes about 4 times more than the baseline (batch size 64 and no gradient alignment regularization). Large and GA take about 10 times more than the baseline. Even though meta-learning takes longer than the baseline, its computation is still less expensive than other alternatives such as large batch size or gradient alignment.
>
> >Could more baselines be included (e.g., the one listed in Sec 6.1).
>
> We would appreciate it if the reviewer can clarify what baselines should be considered.
>
> **Please let us know if our responses do not fully answer your questions or if there are other changes we should make! Thank you!**

---

> ### Author Response · Authors · 2022-08-23
> **Response to Reviewer FBid (1/2)**
>
> Thank you for the reviews and comments.
>
> >How much the proposed metric is different from monitoring of the (squared) td-error (which is the principal metric for any value method)? In other words, wouldn't the conclusion be the same if we take the td-error, say for the sake of simplicity in the deterministic environments?
>
> TD methods do not minimize the squared TD-error; rather they minimize a Bellman error or a projected Bellman error. You are right that in deterministic environments the squared TD error gives the Bellman error, but in general this is not the case. We would measure the Bellman error directly if we could. But, it is hard to do so in stochastic environments. Instead, we justify why we use the difference in TD errors as an approximation in Section 4. We did highlight that this is in fact exact in deterministic settings.
>
> It may seem obvious to use squared TD errors, and we aren't implying this is a highly surprising measure by any means. But, we do try to be careful and point out the metric we really want at different granularities and why TD errors are an ok proxy.
>
> >Comment on the correlation of the Bellman error and value error (in the view of [1])
>
> We would like to first clarify that we do not measure interference for BE as a proxy for interference for value error. Rather, the algorithms are trying to minimize BE, and so we want to know about interference in the objective optimized by the agent. It is relative to the agent, rather than an objective about distance to the true value function.
>
> Let us comment on [1] anyway, and the relationship between BE and VE. [1] show several cases where the Bellman error is not a good proxy for the value error. However, if we have data that follows a distribution that is close to the state-action distribution induced by the policy and an optimal policy, then the Bellman error upper-bounds the value error. In our own experiments, we use reservoir sampling to evaluate the interference. The resulting data distribution should be reasonably close to the distribution of the policy and an optimal policy, so our measurement of BE is actually an ok proxy for VE as well.
>
> >The correlation plot in Section 5
>
> We agree the correlation is not clear without target nets. However, if we look closer, we can see that the agents are unstable and suffer maximal degradation on Acrobot. This failure causes the correlation to be unclear, as we discuss in the text.
>
> Moreover, we don’t expect to see a perfect correlation with Degradation. Interference is about estimated quantities, like values, that represent the agent’s knowledge of the world. Degradation on the other hand cannot be measured by the agent unless the agent has access to a simulator—e.g., it is not readily measurable on a robot or in industrial control settings. Degradation may not immediately occur even with inaccuracies introduced into the value estimates. Further, in some cases, the performance may temporarily be worse due to exploration, even in the absence of interference in the agent’s value estimates.
>
> >Following that, having only two envs in the empirical works feels risky.
>
> We introduce a new methodology to measure interference, to help investigate algorithms across many environments. For most of the sciences, evidence is built up over many papers. It is not expected that one cognitive science lab will run all the child studies needed to conclusively show an outcome. Rather, they add to a growing body of evidence. The critical components are to run clear experiments which can be built on, and in some cases to introduce re-usable methodologies. We cannot conclusively state anything about the behavior of these agents across the many different problems an RL agent could see. This set is vast. Instead, we provide sound evidence (which as you highlight actually took quite a lot of care and compute) in two classic but diverse environments, where RL have been shown to exhibit catastrophic forgetting, and a repeatable methodology for others to continue the work. We had focused on these two environments for the reasons mentioned. But we did also implement and test them in Mountain Car. We can add those plots to the appendix.
>
> >The differences in returns seem not that big in Fig 5 (are they?) Is the gain significant enough to justify the higher computational cost?
>
> Even though the average returns seem close, the interference for the baselines is much higher than the online-aware methods. From figure 6 we can see that the baseline and online-aware method perform very differently in terms of stability, even though the average returns are close.

---

### Review · Reviewer_6kfD · 2022-09-21

**Summary Of Contributions:**

The paper investigates the problem of Catastrophic interference in DRL, which is a common and fundamental problem in DRL. The authors defined an interference evaluation metric as the variance difference of the td-target during learning and empirically found the correlation between this metric and performance degradation. Finally, the authors proposed a practical solution of minimizing this interference metric as a meta-learning problem and showing the effectiveness empirically.


**Requested Changes:**

---the baseline is weak (only using DQI as baseline) without considering directly related benchmarks [1],

---it is better to consider more backbone networks to support the effectiveness and generalization ability of the proposed interference metric.

-- only two games are considered for evaluation and it is not clear why these two games are specifically selected. More game benchmarks should be added to make it more convincing

**Strengths And Weaknesses:**

strength
---the problem is a fundamental question in DRL and has been relatively ignored in the literature.

---the paper provides a formal analysis and introduced a qualitative evaluation metric to measure the interference.

-- the idea of translating the issue to a meta-learning problem is nice and interesting

weakness

---the baseline is weak (only using DQI as baseline) without considering directly related benchmarks [1], Also there are a few works on addressing catastrophic forgetting problems in multi-task or continuous learning in RL, what is the relation and difference with these works?

---it is better to consider more backbone networks to support the effectiveness and generalization ability of the proposed interference metric.

-- only two games are considered for evaluation and it is not clear why these two games are specifically selected. More game benchmarks should be added to make it more convincing

[1] Fuzzy Tiling Activations: A Simple Approach to Learning Sparse Representations Online, ICLR21

---

> ### Author Response · Authors · 2022-09-26
> **Response to Reviewer 6kfD**
>
> Thank you for the reviews and comments.
>
> We would like to first clarify our claims in this paper. We propose a new methodology to measure interference and provide a better understanding of catastrophic interference. We provide evidence that the measure can be used in general environments (for example, it only requires access to a buffer to compute the measure). However, we do not claim that the proposed measure is the best way to measure interference or the online-aware algorithm is the SOTA method to reduce interference.
>
> >There are a few works on addressing catastrophic forgetting problems in multi-task or continuous learning in RL, what is the relation and difference with these works?
>
> For multi-task RL, due to the known task boundary, forgetting can be naturally defined as the performance degradation of one task when we are learning another task. There are several methods that aim to improve generalization or address forgetting in multi-task learning by adding dropout or regularization on the function approximation (Cobbe et al., 2018) or using knowledge distillation. However, in single task or continual RL, it is unclear how to define the task boundary and measure performance degradation in such a case. Therefore, we aim to provide a better understanding of interference and forgetting even in the single task setting. The online-aware algorithm is a natural algorithm to minimize our definition of interference in the single task setting.
>
> >The baseline is weak (only using DQI as baseline) without considering directly related benchmarks [1].
>
> FTA aims to learn a spare representation and a sparse representation can be viewed as a way to implicitly minimize the gradient alignment and hence the update interference. We will add a discussion in the paper. However, it is unclear to us what the reviewer wants us to compare to FTA. We would appreciate it if the reviewer can clarify that.
>
> >Only two games are considered for evaluation and it is not clear why these two games are specifically selected.
>
> We mention in the paper that “we chose Cartpole because RL agents have been shown to exhibit catastrophic forgetting in this environment (Goodrich, 2015). We chose Acrobot because it exhibits different learning dynamics than Cartpole: instead of starting from a good location, it has to explore to reach the goal.” In summary, we chose two environments that have very different learning dynamics and have been shown to exhibit catastrophic forgetting.
>
> >It is better to consider more backbone networks to support the effectiveness and generalization ability of the proposed interference metric. More game benchmarks should be added to make it more convincing
>
> We introduce a new methodology to measure interference, to help investigate algorithms across many environments. The critical components of the paper are to run clear experiments which can be built on, and in some cases to introduce re-usable methodologies. To use experiments to show generality, we need to test it on a significant number of environments with a significant number of network architectures. We cannot conclusively state anything about the behavior of these agents across the many different problems an RL agent could see. This set is vast. Instead, we provide sound evidence in two classic but diverse environments with a standard and common network architecture, and a repeatable methodology for others to continue the work.
>
> **Please let us know if our responses do not fully answer your questions! Thank you!**

---

### Decision · Action_Editors · 2022-11-22

**Recommendation:** Reject

**Comment:**

As explained above the paper is lacking both in terms of theoretical and empirical evidence.  A theoretical analysis of when the proposed notions of interference can be trusted should be added.  In addition, the empirical evaluation should evaluate the impact of the confounding factors acknowledged by the paper.  Finally, additional envronments that are more challenging should be used for the evaluation as well as more advanced baselines than DQI.

**Audience:**

Yes, this work would be of interest to the RL community.

**Claims And Evidence:**

The paper proposes several definitions of interference based on TD updates and suggests that they are correlated with some notion of degradation.  This claim is not supported by any theory.  It is however supported by basic experiments that include two toy problems and one baseline.  However as pointed out by all the reviewers, this is insufficient empirical evidence.  The reviewers also questioned to what extent the proposed notions of interference make sense.  It is intuitive to define interference as the difference in expected squared TD error due to some update, but as the paper acknowledged, there are several confouding factors that can impact expected squared TD error.  However, none of those confounding factors are studied empirically.  In the end, we don't know when we can trust the proposed notions of interference as indicative of degradation.   I also read the paper, so let me make an additional observation.  The paper implicitly assumes that expected squared TD error decreases monotonically unless there is inteference.  However, we can show that expected squarred TD error does not always decrease monotonically even when all the computation is done exactly (i.e., we are doing exact policy evaluation).  Common proofs of converge of exact policy evaluation rely on showing that the Bellman operator is a contraction mapping.  However this contraction mapping does not reduce expected TD error, it reduces the $L_\infty$ distance between two value functions.  Perhaps what is misleading is that the MDP theory uses the TD error to compute an upper bound on the distance to the true value function and this is often used as a stopping criterion, but again expected TD error does not decrease monotonically.  I've had many students try to track the expected TD error durring training in the hope of verifying that their implementation is correct, but it is quite common for the expected TD error to shoot up at the beginning of training before coming down in a non-monotonic way.  A concrete example of this phenomenon arises in domains with sparse rewards.  Suppose that a large positive reward is earned for reaching a terminal goal state and no reward is provided anywhere else.  If we initialize the Q-function to 0, it will have a 0 TD error everywhere except at the states that can reach the terminal goal state.  As we apply the Bellman policy evaluation operator to update the Q function, the expected TD error initially increases since more states can reach those that have non-zero Q-values.  The fact that the proposed notions of interference assume that expected TD error decreases in the absence of interference is a major problem.  That being said, the ideas of the paper are still interesting, but the paper needs to do a careful theoretical and empirical analysis of when the proposed notions of interference can be trusted.